# Observed Antarctic sea ice expansion reproduced in a climate model after correcting biases in sea ice drift velocity

Shantong Sun [1,2✉] & Ian Eisenman [1✉]

The Antarctic sea ice area expanded significantly during 1979–2015. This is at odds with state-of-the-art climate models, which typically simulate a receding Antarctic sea ice cover in response to increasing greenhouse forcing. Here, we investigate the hypothesis that this discrepancy between models and observations occurs due to simulation biases in the sea ice drift velocity. As a control we use the Community Earth System Model (CESM) Large Ensemble, which has 40 realizations of past and future climate change that all undergo Antarctic sea ice retreat during recent decades. We modify CESM to replace the simulated sea ice velocity field with a satellite-derived estimate of the observed sea ice motion, and we simulate 3 realizations of recent climate change. We find that the Antarctic sea ice expands in all 3 of these realizations, with the simulated spatial structure of the expansion bearing resemblance to observations. The results suggest that the reason CESM has failed to capture the observed Antarctic sea ice expansion is due to simulation biases in the sea ice drift velocity, implying that an improved representation of sea ice motion is crucial for more accurate sea ice projections.

[1] Scripps Institution of Oceanography, University of California San Diego, La Jolla, CA, USA. [2] Environmental Science and Engineering, California Institute of Technology, Pasadena, CA, USA. ✉email: shantong@caltech.edu; eisenman@ucsd.edu

Antarctic sea ice expanded during recent decades and then rapidly contracted during the past few years. In this study, we focus on the expansion: during 1979–2015 the Antarctic sea ice area increased at a statistically significant rate that was approximately a third as fast as the Arctic sea ice retreat (Supplementary Fig. 1). This expansion is at odds with basic physical intuition about how sea ice should respond to rising global temperatures, and it is also at odds with state-of-the-art climate models which typically simulate a receding Antarctic sea ice cover in response to climate forcing during this period[1,2].

A number of explanations have been proposed for the enigma that climate models consistently fail to capture the observed Antarctic sea ice expansion. Some studies have focused on the internal variability in Antarctic sea ice simulated by climate models[3–5]. For example, the observed sea ice expansion was shown to be within the range of internal variability of a climate model simulation under constant preindustrial forcing[3]. However, when the sum of the model-simulated internal variability and the model-simulated response to historical greenhouse forcing is considered, the observations fall deep within the tail of the model results[2]. Overall, these studies suggest that although internal variability can give rise to Antarctic sea ice expansion in some cases, a highly unusual realization of internal climate variability would be required to have occurred in the observations for this to explain the observed changes in the Antarctic sea ice.

Alternatively, anthropogenic ozone depletion has been suggested to strengthen the Southern Hemisphere westerly surface winds, leading to an anomalous equatorward Ekman transport that initially causes cooling and sea ice expansion, followed by a slower warming due to upwelling of the warmer deep water[6–8]. Modeling studies have linked the simulated Southern Hemisphere westerly wind to biases in the Antarctic sea ice across different models[9,10]. However, a later study that compared a suite of current climate models with observations suggested that ozone depletion is unlikely to be the primary driver of surface cooling and sea ice expansion in the Southern Ocean[11].

Other explanations have been proposed that also involve changes in surface winds, whether driven by internal variability[12] or ozone depletion[13] or other factors such as greenhouse forcing[14]. Close relationships were found between observational estimates of surface wind, sea ice motion, and sea ice concentration[15]. However, later work focusing on the seasonal structure of regional sea ice trends identified issues with these relationships[16]. Nonetheless, trends in the Southern Ocean winds have been found to be weaker in climate models than in observations[17,18], which has been suggested to influence the sea ice[19].

A number of other mechanisms have also been proposed for the discrepancy between sea ice expansion in observations and sea ice retreat in climate models, including enhanced sea ice growth or diminished melt in the observations due to a stronger ocean stratification caused by warming surface temperatures[20] or an increased meltwater flux from Antarctic glacial discharge[21–23], suppressed warming due to ocean heat uptake[24] or the mean wind-driven upwelling and northward transport of surface waters around Antarctica[25], or sustained internal variability associated with ice-ocean feedbacks[26]. To date, however, the enigma remains unresolved.

Here we investigate the hypothesis that current climate models fail to simulate Antarctic sea ice expansion due to systematic biases in the simulated sea ice drift velocity. We manually correct this bias in a climate model by replacing the simulated sea ice drift with an observational estimate of the sea ice motion field. If biases in the simulated sea ice motion are the main reason that climate models fail to capture the observed Antarctic sea ice expansion, then we expect this correction to substantially improve the simulated Antarctic sea ice changes.

## Results

**Model simulations.** As a control, we use the National Center for Atmospheric Research Community Earth System Model (NCAR CESM) Large Ensemble, which has 40 realizations that all use identical historical and future forcing but differ in their initial conditions[27], referred to here as LENS. These 40 LENS members all undergo Antarctic sea ice retreat during recent decades[2]. In order to test the present hypothesis, we modified CESM to replace the simulated sea ice velocity with an observational estimate of the sea ice motion field (Fig. 1). The observational product was derived from satellite measurements, also drawing on buoy data and NCEP-NCAR reanalysis winds, and it has daily data on a 25-km grid[28]. The simulations with the ice motion specified to follow this observational product are referred to as ObsVi. Further details regarding the model setup and the data product are included in the Methods section.

Although the satellite-derived sea ice motion fields begin in 1979, here we focus on the period 1992–2015 due to issues with the sea ice motion data prior to a satellite sensor transition in December 1991 (see Supplementary Fig. 3). We branch the ObsVi runs from three separate realizations of recent climate change (LENS-2, LENS-4, and LENS-6, using the indices associated with each run in the LENS archive). This leads to three simulations with sea ice motion specified from the observed time-varying field (ObsVi-2, ObsVi-4, and ObsVi-6). We focus on the annual-mean sea ice area.

**Antarctic sea ice changes.** The Antarctic sea ice expands in all three ObsVi runs, with one run having ice expansion at a rate similar to the observed value of $33 \times 10^3 \, \text{km}^2$ per year (Fig. 2a). This is in contrast to the three LENS control runs, which all have Antarctic sea ice retreat at a rate faster than $-29 \times 10^3 \, \text{km}^2$ per year. We emphasize that the only difference between the two sets of runs is in the sea ice motion field.

Figure 2b indicates that the runs with observed ice motion (ObsVi) all lie outside the range of what CESM allows with simulated ice motion (LENS). All 40 of the LENS runs undergo varying levels of Antarctic sea ice retreat, whereas the three ObsVi runs all undergo expansion.

We illustrate the spatial structure of the sea ice changes using the meridionally-integrated sea ice area trend, i.e., the linear trend in the annual-mean sea ice concentration integrated over each longitudinal sector (Fig. 3). The observed sea ice cover expands at almost every longitude, except for relatively small parts of the western Pacific and the Amundsen-Bellingshausen Sea, where the values are slightly negative. Note that this observed zonal structure in sea ice expansion during 1992–2015 is somewhat different from the trend calculated over longer periods such as 1979–2015 (Supplementary Fig. 5), which shows substantial sea ice retreat in the Amundsen-Bellingshausen Sea as discussed in some previous studies[29].

The spatial structure of the sea ice trend in the LENS control runs does not resemble the observed trend (Fig. 3a). At nearly all longitudes, at least 2 of the 3 runs have a receding sea ice cover.

The simulated spatial structure of the expansion in the ObsVi runs, however, bears resemblance to the observations (Fig. 3b). Most of the sea ice expansion in the ObsVi runs takes place at the Ross Sea, Weddell Sea, and the western Indian sector of the Southern Ocean, as in the observations. Note that the pronounced expansion in these regions is partly compensated by the sea ice retreat in the eastern Indian and western Pacific sectors, where there is a shift in the trend compared with the observations. The spread among the ObsVi runs in Fig. 3b is narrower than the LENS runs in Fig. 3a, particularly in the Amundsen-Bellingshausen and Weddell Seas where there is substantial internal climate variability[30–32]. Quantitatively, the zonal

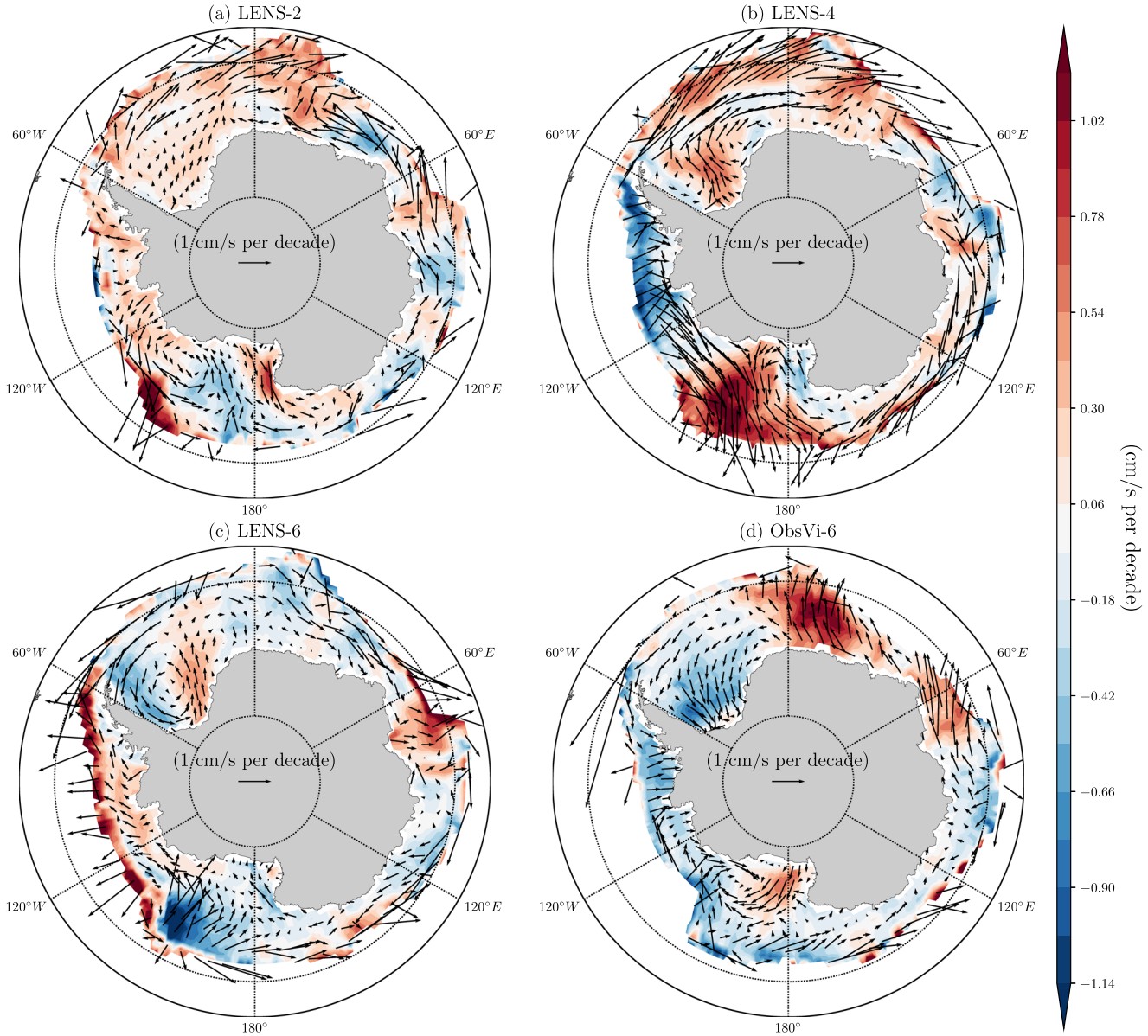

**Fig. 1 Antarctic sea ice drift velocity trend.** Linear trend in the annual-mean sea ice drift velocity (vector) during 1992–2015, with the linear trend in the meridional velocity component also indicated (shading), in **a** LENS-2, **b** LENS-4, **c** LENS-6, and **d** ObsVi-6. Note that the sea ice velocity trends in the observations, ObsVi-2, and ObsVi-4 are approximately equivalent to ObsVi-6 (see Supplementary Fig. 2).

average of the difference between the highest and lowest plotted values is 159 km$^2$/year/deg in Fig. 3b and 170 km$^2$/year/deg in Fig. 3a. This implies that sea ice motion exerts a relatively strong control on the spatial structure of the sea ice area changes.

Taken together, these results suggest that the reason CESM fails to simulate the observed Antarctic sea ice expansion is due to simulation biases in the sea ice drift velocity.

## Discussion

The key factors that determine the sea ice drift velocity in CESM include surface winds, ocean surface currents, sea ice rheology, and sea ice drag coefficients. We carried out an additional set of simulations to test the importance of biases in the simulated surface winds influencing the sea ice drift. In these runs (referred to as ERAWind), we replaced the simulated surface wind in the sea ice momentum calculation with ERA-Interim[33] reanalysis wind vectors. The ERAWind runs have a slower sea ice retreat

than the LENS runs (Supplementary Fig. 6), but the ice does not expand like in the ObsVi runs, implying that surface wind biases may be partially responsible for the relevant biases in the simulated sea ice drift velocity. The spatial structure of the sea ice trend in the ERAWind runs (Supplementary Fig. 7) bears a level of resemblance to the observations that is broadly similar to that of the ObsVi runs (Fig. 3b).

We investigated the relationship between sea ice drift velocity and sea ice area in the simulations, and we found no clear connection between the trends (see Supplementary Figs. 8 and 9 and Budget analysis section in the Supplemental material). The ice expansion may possibly be attributable to an increased northward drift velocity, but this relationship is not straightforward and varies by region and season (Supplementary Fig. 8). Despite the sea ice area trend in the ObsVi runs falling outside of the range of the LENS results (Fig. 2b), we found no clear systematic bias in the sea ice velocity trend in the LENS runs (Fig. 1). For example, in LENS-4 there is a substantial increase in both northward

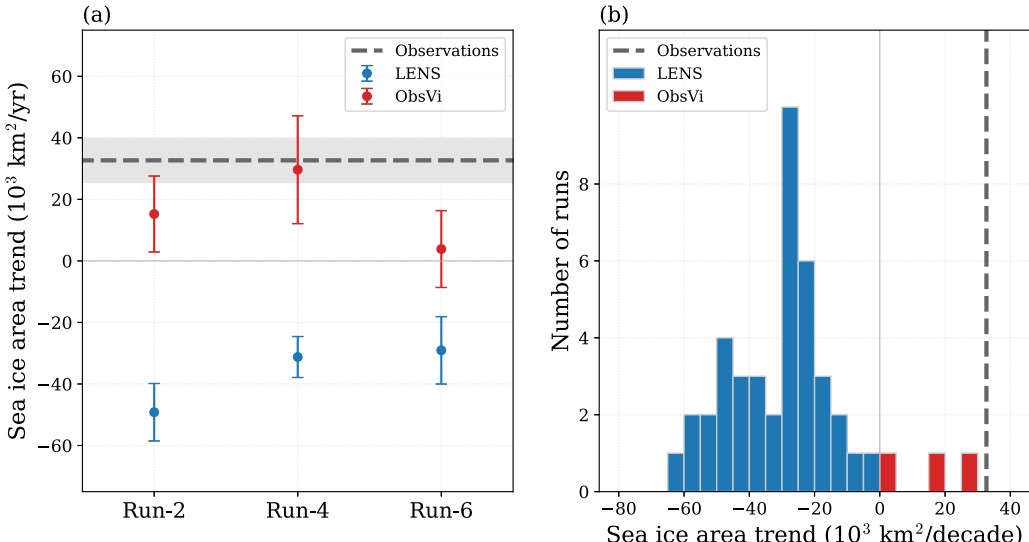

**Fig. 2 Antarctic sea ice area trend. a** Linear trend in the annual-mean sea ice area during 1992–2015 in the observations (gray dashed line) and the LENS (blue dots) and ObsVi (red dots) simulations. The error bars show the standard error associated with the linear trends, which are calculated using ordinary least squares regression. **b** Histogram of the linear trends in annual-mean sea ice area for the 40 CESM LENS runs (blue) and the three ObsVi runs (red), along with the linear trend in the observations (gray dashed line). Note that the result is approximately equivalent when ice extent is used rather than ice area (see Supplementary Fig. 4).

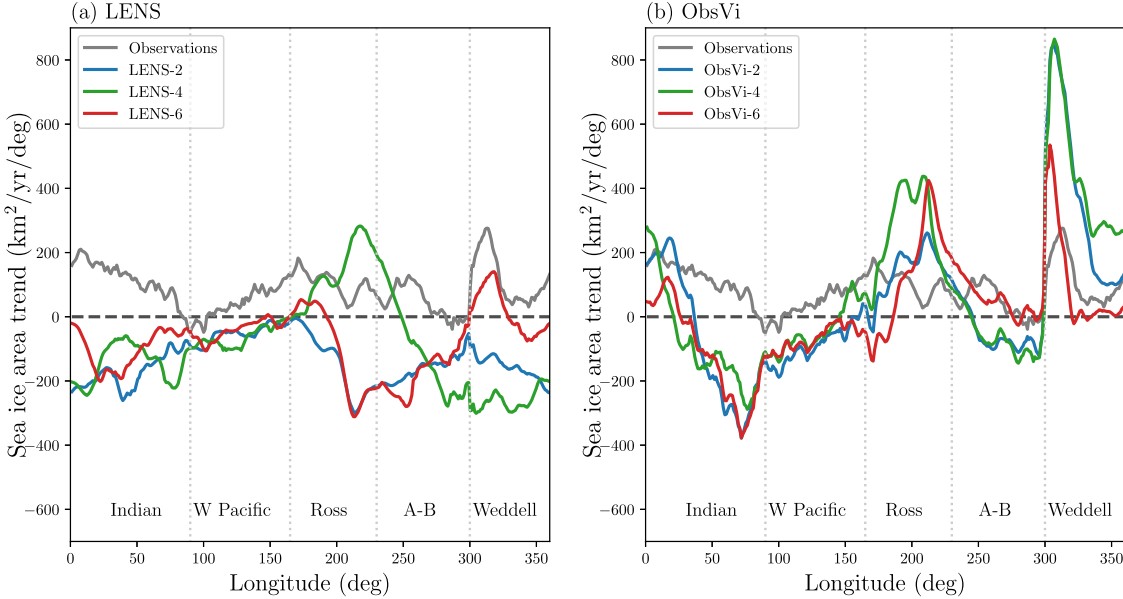

**Fig. 3 Spatial structure of Antarctic sea ice trend.** Linear trend in the annual-mean meridionally-integrated sea ice area during 1992–2015 as a function of longitude in the **a** LENS and **b** ObsVi simulations. Observations are plotted for comparison as a gray line in both panels. The longitude ranges of the different Southern Ocean sectors are labeled and separated with gray dotted lines. Here, "A-B" stands for the Amundsen-Bellingshausen Sea and "W Pacific" refers to the western Pacific Ocean.

sea ice motion in the Ross Sea and southward sea ice motion in the Amundsen-Bellingshausen Sea; this is much weaker in observations, and the trends are opposite in LENS-2 and LENS-6. In contrast to the ice velocity trend, there do appear to be noteworthy biases in the mean state of the ice velocity (Supplementary Fig. 10), which may plausibly play a role in setting the ice area trends.

Several important caveats should be emphasized. (i) The use of just three ObsVi ensemble members may be insufficient to resolve the influence of sea ice motion biases on the sea ice trend in CESM due to internal variability. (ii) Despite substantial improvement, there are still notable differences between the observations and the ObsVi runs in terms of the spatial structure of the changes (Fig. 3). (iii) These results do not resolve what specific features of the biases in the simulated sea ice velocity field are most important for the sea ice area trend. (iv) Questions remain regarding the physical mechanism by which the sea ice velocity field influences the sea ice area in these simulations. (v) We can not rule out the possibility that the simulations with specified ice velocity are producing realistic sea ice area trends for the wrong reasons due to cancellation of errors in the simulation results. (vi) Relatedly, there may be substantial errors in the observationally-based ice velocity fields that we use to specify the ice motion.

In conclusion, like most current climate models, CESM does not simulate the observed Antarctic sea ice expansion. These results show that this can be improved by manually correcting sea ice drift velocity biases. Some of this improvement can be captured by instead correcting biases in the surface winds in the sea ice momentum equation. The main candidates for explaining the remainder of the discrepancy between simulated and observed sea ice changes include model biases in the sea ice rheology, sea ice drag coefficients, and ocean surface currents, as well as ice velocity biases due to the coarse model resolution. Our results suggest that an improved representation of sea ice motion is crucial for more accurate sea ice projections.

## Methods

**Satellite-derived data**. We use the Polar Pathfinder Daily Sea Ice Motion Vectors[28], which is managed by the National Snow & Ice Data Center (NSIDC). This dataset includes sea ice velocity fields for both hemispheres, which are derived from satellite measurements and also draw on buoy measurements as well as free drift estimates calculated from NCEP-NCAR reanalysis geostrophic winds. It provides sea ice velocities that are interpolated onto a 25-km resolution Equal Area Scalable Earth (EASE) grid with daily temporal resolution from October 1978 to January 2016 at the time the data were downloaded. Here we use data during 1992–2015. As discussed above, we omit the earlier years due to data issues prior to a December 1991 satellite sensor transition (Supplementary Fig. 3), and we truncate the end of the dataset because we focus on the period of sea ice expansion. We interpolate the ice drift velocity from the 25-km resolution EASE grid to the nominal 1° resolution CESM model grid by averaging the observations with grid centers that are located within each model grid cell.

For the observed sea ice concentration, we use the monthly-mean Sea Ice Concentrations from Nimbus-7 SMMR and DMSP SSM/I-SSMIS Passive Microwave Data[34], which is generated using the NASA Team algorithm from brightness temperature data based on multiple sensors including the Nimbus-7 SMMR, the Defense Meteorological Satellite Program (DMSP)-F8, -F11, and -F13 SSM/I, and the DMSP-F17 Special Sensor Microwave Imager/Sounder (SSMIS). The ice concentration is provided on a 25-km resolution polar stereographic grid. We use the NSIDC Sea Ice Index[35] for the observed ice area time series, as well as for the observed ice extent time series plotted in Supplementary Figs. 1b and 4 (note that the ice extent is defined as the total area of grid boxes with sea ice concentration >15%).

We use reanalysis surface winds from ERA-Interim[33], which has been suggested to provide a somewhat reliable estimate for the Southern Ocean surface fields[36,37]. The reanalysis goes back to 1979, and we use wind data during 1979–2015. The wind product is reported on a 0.75° grid resolution with 6-h frequency. We interpolate it to the model grid using bilinear interpolation.

**Model setup**. In the ObsVi runs, the sea ice momentum equation is replaced with a relaxation to the satellite-derived ice velocity field,

$$\frac{d\vec{v}}{dt} = \frac{1}{\tau}(\vec{v}_{obs} - \vec{v}), \qquad (1)$$

where $\vec{v}$ represents the sea ice drift velocity in the model, and $\vec{v}_{obs}$ denotes the daily specified sea ice drift velocity. We choose a short restoring timescale $\tau = 1$ h to constrain the sea ice drift velocity to resemble observations. The momentum equation is sub-cycled during each sea ice model time step (using the CESM parameter xndt_dyn) in order to avoid numerical instability. In locations where the satellite-retrieved sea ice velocity data is not available but there is simulated ice, we use the ice momentum equation with ERA-Interim surface winds as in the ERAWind runs. In the ERAWind runs, the default ice momentum equation is used but the surface wind used to generate the atmosphere-ice stress is replaced with ERA-Interim winds; note that the model wind field is altered only in the calculation of the atmosphere-ice stress in the sea ice momentum equation.

**Spinup of simulations**. The three ObsVi runs are branched from the corresponding LENS runs on January 1, 1960. For each ObsVi run, we spin-up the model during simulation years 1960–1991 by relaxing the sea ice velocity to the observed mean annual cycle (averaged over 1992–2015), and then the simulation is continued during 1992–2015 using the full time evolution of the observed ice motion field. In other words, we computed the mean annual cycle in the daily observational field, and the ice motion is relaxed to this field every year during 1960–1991, although increases in greenhouse gas forcing and other forcing changes during this period are equivalent to the LENS runs.

Due to the change in sea ice momentum forcing in 1960, the Antarctic sea ice area increases rapidly during the first few months such that the annual-mean ice area increases by around $1 \times 10^6$ km$^2$ in the first year (blue lines in Supplementary Fig. 11a). The ice area then declines for a decade or so and then remains relatively

constant during the following decade or so. After the 1960–1991 spin-up period, the ObsVi runs gain sea ice during the 1992–2015 analysis period.

The three ERAWind runs are similarly branched from the corresponding LENS runs on January 1, 1960. Since the ERA-Interim winds are available for a longer time period, we spin-up the ERAWind runs with the 1979 forcing repeating every year during simulation years 1960–1978 and then use the full time evolution of the wind field during 1979–2015, thereby allowing further spin-up during 1979–1991 before the 1992–2015 analysis period.

We find that the Antarctic sea ice area also initially increases in the ERAWind runs (blue lines in Supplementary Fig. 11b). This initial increase is smaller than in the ObsVi runs, and the ice area in the ERAWind runs remains relatively close to the LENS runs throughout the simulation period.

Additional simulations to investigate the sensitivity to spin-up conditions are presented in the Supplemental material (Sensitivity of simulations to spinup conditions section).

**Year-to-year variability**. Although the 1992–2015 ice area trends agree better with observations in the ObsVi runs than in the ERAWind runs, the ERAWind runs show better agreement with observed year-to-year changes in the ice area. This is listed in Supplementary Table 1. The correlations with observations for the detrended annual-mean ice area during 1992–2015 range from 0.38 to 0.62 in the three ERAWind runs. Note that the supplementary runs that use different spin-up conditions (ERAWind_1992Spinup and ERAWind_ClimSpinup described in the Supplemental material) have fairly similar correlations. This implies that despite not capturing as much of the long-term trend, the ERAWind runs may capture more of the observed year-to-year variability. A concurrent study using CESM simulations shows a similar result: when the wind field is nudged toward ERA-interim, the model captures much of the observed year-to-year variability in Antarctic sea ice extent[38].

**Seasonal variations**. Although this study focuses on annual-mean trends, some previous studies have examined seasonal variations in the observed trends[16,39,40]. We find that the seasonal structure of the 1992–2015 ice area trend varies considerably between the three ObsVi runs (Supplementary Fig. 12), without a consistent structure in the bias with the observations.

**Arctic sea ice trends**. The changes in the ice momentum equation in all of the runs in this study apply in both hemispheres. However, in the Arctic we do not find that any of these changes lead to substantially more accurate simulations of the sea ice area trend or the year-to-year variability (Supplementary Table 1).

## Data availability
Model simulation output fields that support the findings of this study are available in figshare at https://doi.org/10.6084/m9.figshare.12857672.

## Code availability
The CESM code modifications used in this study can be accessed at https://stsun.github.io/files/Sun-Eisenman-CESMCode-2020.tar.

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

## Acknowledgements

Without implying their endorsement, we thank Ed Blanchard-Wrigglesworth, Till J.W. Wagner, Paul Holland, and Elizabeth Hunke for helpful comments and discussions. This work was supported by National Science Foundation Grant OPP-1643445.

## Author contributions

I.E. and S.S. designed the simulations and analysis, S.S. carried out the simulations and analysis, and S.S. and I.E. wrote the manuscript.

## Competing interests

The authors declare no competing interests.
