## [Peer Review File · Nature Communications]

REVIEWER COMMENTS

Reviewer #1 (Remarks to the Author):

Review of Sun and Eisenman:

Overall, I find the paper well-written and the results interesting. Although the paper does not address why the modelled and observed ice drift velocities differ, the modeling experiments effectively isolate the drift velocities as opposed to the surface winds (assuming the reanalysis is adequate). I recommend this paper for publication after minor revisions.

I have a couple of questions about the methodology. First, when the observed drift velocities are being applied to the model, what happens if there is ice in the model grid cell but no drift velocities in the observations and vice versa? It is not clear from the Methods section how these cases are handled. I would appreciate it if the authors could expand on this.

Second, I'm curious to know to what extent the ObsVi sea ice climatology has been altered before the 1992-2015 time period (due to the spin-up) and to what extent this spin-up affects the results. What happens if the spin-up time is shortened? What is the maximum spin-up time required to achieve a good match between the ObsVi runs and the observations for 1992-2015? I would like to see a supplementary figure showing the difference in the sea ice climatology between the LENS and ObsVi runs for the spin-up time, 1960-1991 to the reader a sense of this.

I have a few suggestions to make the figure captions a bit clearer:

Figure 1: Is this just for the 1992-2015 time period? If so, add this to the caption.

Figure 3: Again, add 1992-2015 to the caption.

Figure S1: Does this plot show data from 1979-2019? This is not in the figure caption.

Figure S3: 1992-2015?

Figure S6: This is observational sea ice area, correct? This is not clear from the caption.

Figure S8: 1992-2015? Also, I think you mean ObsVi-6, not ObsVi-2 in the caption.

- Karen Smith

Reviewer #2 (Remarks to the Author):

The submission has the potential to make a significant contribution to the literature, but it is not quite there yet. Before I would be able to recommend acceptance, there are a number of key issues which need to be addressed.

Lines 13-17: I have a little trouble as to interpretation. Specifying the OBSERVED ice drift in the model results in model simulations which show increases in Antarctic sea ice extent (in agreement with the observations). By this specification, to what extent are we feeding in the answer to how the ice should behave? If dynamic processes are the dominant driver the sea ice distribution then the model is essentially forced (or nudged, with an e-folding time of 1 hour) to follow the observations. However, if thermodynamic processes are of greater weight, one might have expected the model to behave somewhat differently. The relative importance of these two mechanisms has recently been shown to give 'fast' and 'slow' responses of Southern Ocean (see, e.g.,

Yavor Kostov, John Marshall, Ute Hausmann, Kyle C. Armour, David Ferreira and Marika M. Holland, 2017: Fast and slow responses of Southern Ocean sea surface temperature to SAM in coupled climate models. *Climate Dynamics*, 48, 1595-1609, doi: 10.1007/s00382-016-3162-z.

Marika M. Holland, Laura Landrum, Yavor Kostov and John Marshall, 2017: Sensitivity of Antarctic sea ice to the Southern Annular Mode in coupled climate models. *Climate Dynamics*, 49, 1813-1831, doi: 10.1007/s00382-016-3424-9.

I would like to see a bit more critical discussion on the experimental design, and how the results are to be interpreted.

Lines 23-25: Fig. S1 shows only the ANNUAL-MEAN sea ice extents in the two polar regions. The MONTHLY trends reveal considerably more structure. In this context very valuable to refer to the analysis of Simmonds, 2015: Comparing and contrasting the behaviour of Arctic and Antarctic sea ice over the 35-year period 1979-2013. *Ann. Glaciol.*, 56, 18-28.

Lines 30-64: Important to add the consequences of errors in the simulation of the SH jet in CMIP5 models. It has been shown that jet errors are highly correlated with biases in simulated Antarctic sea ice. The authors should include this point in the Introduction, and make reference to the paper of Screen & co-authors (2018). Polar climate change as manifest in atmospheric circulation. *Current Climate Change Reports*, 4, 383-395. doi: 10.1007/s40641-018-0111-4

Lines 125-127: The authors state that 'The spread among the ObsVi runs in Figure 3b is narrower than the LENS runs in Figure 3a, implying that the sea ice motion exerts a strong control on the spatial structure of the sea ice area changes.' This seems to be the only comment made in connection with Fig. 3b, but it misses perhaps the most informative aspect of the Fig. Fig. 3b shows that about half the longitudinal range shows negative trends in the three ObsVi runs in the eastern hemisphere, and for much of that domain the negative (erroneous) trends appear to be LARGER than those in the three LENS runs.

It needs to be pointed out that the APPARENT massive improvement in the trend of the annual mean of the total sea ice area (Fig. 2a) in fact comes about from the compensation of large positive and negative errors (Fig. 3b). Hence this result is much less impressive than is implicitly claimed here.

I am puzzled by the lack of similarity between Fig. 3b and the red line (which I take to be the mean of the three ObsVi ensemble members) in Fig. S6. Please to clarify here.

As a final point here, the three realisations Obsvi- (2, 4, and 6) show fairly similar trends (as the authors note), but with notable exceptions in the Amundsen-Bellingshausen and Weddell Seas. Valuable to add a few words as to what might be going on here. These locations are key nodes of the Pacific-South American teleconnection pattern and, because of the PSA association with El Nino undergo considerable variability. In this additional comment the authors should refer to the analysis of Irving et al. (2016) A new method for identifying the Pacific-South American pattern and its influence on regional climate variability. *J. Climate*, 29, 6109-6125. In addition the ABS is a key location for SH wavenumber 3 (Irving & ... 2015 - A novel approach to diagnosing Southern Hemisphere planetary wave activity and its influence on regional climate variability. *J. Clim.*, 28, 9041-9057), as well as being strongly influence by synoptic variations (make reference to Fogt, Wovrosh and co-authors (2012) The characteristic variability and connection to the underlying synoptic activity of the Amundsen-Bellingshausen Seas Low. *J. Geophys. Res.*, 117, D07111, doi: 10.1029/2011JD017337. It will be very helpful to the interpretation here to comment on the above points.

Lines 165-172: The paper presents some appropriate caveats associated with the study. Valuable to also include caveats based on the comments made above.

Lines 174-176: In the conclusions the authors state '... most current climate models ... [do] not simulate the observed Antarctic sea ice expansion. Our results show that this can be fixed by manually correcting sea ice drift velocity biases' In light of my comments above I don't think the second sentence here is an appropriate conclusion to be drawn from the results. Please rephrase this to reflect what has actually been demonstrated.

Please check the References more carefully. Numerous ones have incomplete or erroneous bibliographic details. Some I notice were ...

Lines 223-224:

Kwok, R., 2011: Observational assessment of Arctic Ocean sea ice motion, export, and thickness

in CMIP3 climate simulations. *J. Geophys. Res.*, 116, C00D05, doi: 10.1029/2011JC007004.

Lines 228-230:

Pauling, A. G., I. J. Smith, P. J. Langhorne and C. M. Bitz, 2017: Time-dependent freshwater input from ice shelves: Impacts on Antarctic sea ice and the Southern Ocean in an Earth System Model. *Geophys. Res. Lett.*, 44, 10454-10461, doi: 10.1002/2017GL075017.

Lines 246-247:

Pauling, A. G., I. J. Smith, P. J. Langhorne and C. M. Bitz, 2017: Time-dependent freshwater input from ice shelves: Impacts on Antarctic sea ice and the Southern Ocean in an Earth System Model. *Geophys. Res. Lett.*, 44, 10454-10461, doi: 10.1002/2017GL075017.

Lines 246-247:

Singh H. A., Polvani L. M. and Rasch P. J. (2019) Antarctic sea ice expansion, driven by internal variability, in the presence of increasing atmospheric CO₂. *Geophys. Res. Lett.* 46, 14762-14771, doi: 10.1029/2019GL083758.

Responses to Reviewer #1

We thank the reviewer for these helpful comments. Based on these comments, we have revised the manuscript, including clarifying a number of points that the reviewer identified as unclear or incomplete. We have addressed the comments as follows in the revised manuscript (reviewer’s comments in italics):

[1] I have a couple of questions about the methodology. First, when the observed drift velocities are being applied to the model, what happens if there is ice in the model grid cell but no drift velocities in the observations and vice versa? It is not clear from the Methods section how these cases are handled. I would appreciate it if the authors could expand on this.

Reply: We appreciate the reviewer bringing up that this point was unclear in the original manuscript. In response to this comment, we added the following discussion in the revised supporting information:

In places where the satellite-retrieved sea ice velocity data are not available but there is simulated ice, we use the default momentum equation to calculate the sea ice velocity but with the surface wind replaced with the ERA-Interim reanalysis data.

Note that in locations with observed ice velocities (because there is observed ice) but no simulated ice cover, nothing needs to be done.

[2] Second, I’m curious to know to what extent the ObsVi sea ice climatology has been altered before the 1992-2015 time period (due to the spin-up) and to what extent this spin-up affects the results. What happens if the spin-up time is shortened? What is the maximum spin-up time required to achieve a good match between the ObsVi runs and the observations for 1992-2015? I would like to see a supplementary figure showing the difference in the sea ice climatology between the LENS and ObsVi runs for the spin-up time, 1960-1991 to the reader a sense of this.

Reply: In the revised supporting information, we include a new figure (Figure S5) that shows the Antarctic sea ice area evolution during 1960-2015 for the CESM simulations described in the manuscript. The results in Figure S5 suggest a likely spin-up time of around 20 years, during which the Antarctic sea ice area decreases in ObsViClim (and equivalently ObsVi). In the revised supporting information, we briefly discuss this as follows: “Note that due to the change in sea ice momentum forcing at 1960, the ObsVi and ObsViClim runs experience substantial Antarctic sea ice loss during the first 20 years of the spin-up process, whereas ERAWind does not show a substantial change in trend during 1960-1991 (Figure S5).”

[3] Figure 1: Is this just for the 1992-2015 time period? If so, add this to the caption.

Reply: We appreciate the reviewer catching this omission. Added as suggested.

[4] Figure 3: Again, add 1992-2015 to the caption.

Reply: Also added as suggested.

[5] Figure S1: Does this plot show data from 1979-2019? This is not in the figure caption.

Reply: Yes, this plot shows data from 1979-2019. We appreciate the reviewer catching this omission, and this time range has been added to the revised figure caption.

[6] *Figure S3: 1992-2015?*

Reply: Added as suggested.

[7] *Figure S6: This is observational sea ice area, correct? This is not clear from the caption.*

Reply: Correct, and we appreciate the reviewer catching this omission. This has been clarified in the revised caption, which now reads, “Comparison of the linear trend in observed annual-mean meridionally-integrated sea ice area calculated over 1992-2015 (blue) with that calculated over 1979-2015 (red).”

[8] *Figure S8: 1992-2015? Also, I think you mean ObsVi-6, not ObsVi-2 in the caption.*

Reply: We are grateful to the reviewer for catching this typo. The caption has been modified as suggested in the revised manuscript.

Responses to Reviewer #2

We thank the reviewer for these helpful comments. Based on these comments, we have revised the manuscript, including adding discussions regarding the papers that the reviewer suggested. We have addressed the comments as follows in the revised manuscript (reviewer's comments in italics):

*[1] Lines 13-17: I have a little trouble as to interpretation. Specifying the OBSERVED ice drift in the model results in model simulations which show increases in Antarctic sea ice extent (in agreement with the observations). By this specification, to what extent are we feeding in the answer to how the ice should behave? If dynamic processes are the dominant driver the sea ice distribution then the model is essentially forced (or nudged, with an e-folding time of 1 hour) to follow the observations. However, if thermodynamic processes are of greater weight, one might have expected the model to behave somewhat differently. The relative importance of these two mechanisms has recently been shown to give 'fast' and 'slow' responses of Southern Ocean (see, e.g., Yavor Kostov, John Marshall, Ute Hausmann, Kyle C. Armour, David Ferreira and Marika M. Holland, 2017: Fast and slow responses of Southern Ocean sea surface temperature to SAM in coupled climate models. *Climate Dynamics*, 48, 1595-1609, doi: 10.1007/s00382-016-3162-z. Marika M. Holland, Laura Landrum, Yavor Kostov and John Marshall, 2017: Sensitivity of Antarctic sea ice to the Southern Annular Mode in coupled climate models. *Climate Dynamics*, 49, 1813-1831, doi: 10.1007/s00382-016-3424-9. I would like to see a bit more critical discussion on the experimental design, and how the results are to be interpreted.*

Reply: This is essentially the central question of this manuscript (How much does the ice drift dictate the ice extent in the observed Antarctic sea ice expansion?). As the reviewer correctly indicates, if the dynamic processes are dominant in the observed Antarctic sea ice expansion, the specified relaxation of sea ice velocity to observations would be expected to constrain the simulated Antarctic sea ice change to be similar to observations. However, if other processes such as the thermodynamic processes are dominant in the observed Antarctic sea ice expansion, this relaxation of sea ice velocity would not have substantial impacts on the simulated Antarctic sea ice changes. We now explain this more explicitly in the revised introduction, which reads, "In this study, we manually correct this bias in a climate model by replacing the simulated sea ice drift with an observational estimate of the sea ice motion field. If biases in the simulated sea ice motion are the main reason that climate models fail to capture the observed Antarctic sea ice expansion, then we expect this correction to substantially improve the simulated Antarctic sea ice changes." (Lines 74-79). The results of our simulations are consistent with the hypothesis that sea ice motion is crucial for the observed Antarctic sea ice changes, suggesting the importance of dynamic processes. Nevertheless, we also note that the connection between ice motion and ice extent is subtle and complex rather than straightforward, as can be readily seen for example in our simulation results.

In the revised manuscript, we have also noted the fast and slow responses of the Southern Ocean to a stronger westerly wind, as suggested: "Alternatively, anthropogenic ozone depletion has been suggested to strengthen the Southern Hemisphere westerly surface winds, leading to an anomalous equatorward Ekman transport that initially causes cooling and sea ice expansion, followed by a slower warming due to upwelling of the warmer deep water (e.g., Ferreira et al., 2015; Kostov et al., 2017; Holland et al., 2017)." (Lines 50-54)

[2] Lines 23-25: *Fig. S1 shows only the ANNUAL-MEAN sea ice extents in the two polar regions. The MONTHLY trends reveal considerably more structure. In this context very valuable to refer to the analysis of Simmonds, 2015: Comparing and contrasting the behaviour of Arctic and Antarctic sea ice over the 35-year period 1979-2013. Ann. Glaciol., 56, 18-28.*

Reply: In this study, we have focused on the annual-mean sea ice changes. In response to this comment, we added mention of the seasonal dependence in sea ice changes to the revised manuscripts, which reads, “Throughout this study, we focus on the annual-mean sea ice area, although it should be noted that there are seasonal variations in the observed trends (e.g., Eisenman et al., 2014; Simmonds, 2015).” (Lines 103-105)

[3] Lines 30-64: *Important to add the consequences of errors in the simulation of the SH jet in CMIP5 models. It has been shown that jet errors are highly correlated with biases in simulated Antarctic sea ice. The authors should include this point in the Introduction, and make reference to the paper of Screen & co-authors (2018). Polar climate change as manifest in atmospheric circulation. Current Climate Change Reports, 4, 383-395. doi: 10.1007/s40641-018-0111-4*

Reply: We appreciate the reviewer raising this point, and we have added a reference to the consequences of errors in the simulated SH jet in the revised manuscript, which reads, “Modeling studies have linked the simulated Southern Hemisphere westerly wind to biases in the Antarctic sea ice across different models (e.g., Bracegirdle et al., 2018; Screen et al., 2018).” (Lines 54-56)

[4] Lines 125-127: *The authors state that ‘The spread among the ObsVi runs in Figure 3b is narrower than the LENS runs in Figure 3a, implying that the sea ice motion exerts a strong control on the spatial structure of the sea ice area changes.’ This seems to be the only comment made in connection with Fig. 3b, but it misses perhaps the most informative aspect of the Fig. 3b. Fig. 3b shows that about half the longitudinal range shows negative trends in the three ObsVi runs in the eastern hemisphere, and for much of that domain the negative (erroneous) trends appear to be LARGER than those in the three LENS runs. It needs to be pointed out that the APPARENT massive improvement in the trend of the annual mean of the total sea ice area (Fig. 2a) in fact comes about from the compensation of large positive and negative errors (Fig. 3b). Hence this result is much less impressive than is implicitly claimed here.*

Reply: This is a fair point. In response to this comment, we had added more detailed discussions on the spatial structure of sea ice changes in the revised manuscript (Lines 139-148):

The simulated spatial structure of the expansion in the ObsVi runs, however, bears resemblance to observations (Figure 3b). Most of the sea ice expansion in the ObsVi runs takes place at the Ross Sea, Weddell Sea, and the western Indian sectors of the Southern Ocean, as in the observations. Note that the pronounced expansion in these regions are partly compensated by the sea ice retreat in the eastern Indian and western Pacific sectors, where there is a shift in the trend compared with the observations. The spread among the ObsVi runs in Figure 3b is narrower than the LENS runs in Figure 3a, particularly in the Western Antarctica where there is strong internal climate variability (e.g., Fogt et al., 2012; Irving and Simmonds, 2015, 2016), implying that the sea ice motion exerts a strong control on the spatial structure of the sea ice area changes.

[5] *I am puzzled by the lack of similarity between Fig. 3b and the red line (which I take to be the mean of the three ObsVi ensemble members) in Fig. S6. Please to clarify here.*

Reply: We appreciate the reviewer noting that this was not clear in the original manuscript. The red line in Figure S6 (now Figure S7) shows the linear trend in annual-mean meridionally-integrated sea ice area calculated over 1979-2015 in observations. This has been clarified in the revised caption of Figure S7.

[6] *As a final point here, the three realisations Obsvi- (2, 4, and 6) show fairly similar trends (as the authors note), but with notable exceptions in the Amundsen-Bellingshausen and Weddell Seas. Valuable to add a few words as to what might be going on here. These locations are key nodes of the Pacific-South American teleconnection pattern and, because of the PSA association with El Nino undergo considerable variability. In this additional comment the authors should refer to the analysis of Irving et al. (2016) A new method for identifying the Pacific-South American pattern and its influence on regional climate variability. J. Climate, 29, 6109–6125. In addition the ABS is a key location for SH wavenumber 3 (Irving & ... 2015 - A novel approach to diagnosing Southern Hemisphere planetary wave activity and its influence on regional climate variability. J. Clim., 28, 9041-9057), as well as being strongly influence by synoptic variations (make reference to Fogt, Wovrosh and co-authors (2012) The characteristic variability and connection to the underlying synoptic activity of the Amundsen-Bellingshausen Seas Low. J. Geophys. Res., 117, D07111, doi: 10.1029/2011JD017337. It will be very helpful to the interpretation here to comment on the above points.*

Reply: We appreciate these helpful suggestions.

The ObsVi runs indeed show somewhat more inter-model spread in the Amundsen-Bellingshausen and Weddell Seas than other regions, but with the sea ice velocity constrained by observations, it is not straightforward how the mechanisms the reviewer mentions are related to the inter-model spread in ObsVi. However, the LENS runs have substantially larger inter-model spread than the ObsVi runs in these regions, and this could plausibly be due to the mechanisms mentioned by the reviewer. We have noted this in the revised manuscripts, which now reads, “The spread among the ObsVi runs in Figure 3b is narrower than the LENS runs in Figure 3a, particularly in the Western Antarctica where there is strong internal climate variability (e.g., Fogt et al., 2012; Irving and Simmonds, 2015, 2016), implying that the sea ice motion exerts a strong control on the spatial structure of the sea ice area changes.” (Lines 144-148)

[7] *Lines 165-172: The paper presents some appropriate caveats associated with the study. Valuable to also include caveats based on the comments made above.*

Reply: Agreed. In response to this comment, we have added the following caveat in the revised discussion: “Despite substantial improvement in the simulated Antarctic sea ice trends, there are still notable differences between the ObsVi simulations and the observations in terms of the spatial structure of the changes (Figure S9).” (Lines 182-185)

[8] *Lines 174-176: In the conclusions the authors state ‘... most current climate models ... [do] not simulate the observed Antarctic sea ice expansion. Our results show that this can be fixed by manually correcting sea ice drift velocity biases’ In light of my comments above I don’t think the second sentence here is an appropriate conclusion to be drawn from the results. Please rephrase this to reflect what has actually been demonstrated.*

Reply: Agreed. In the revised manuscript, this has been rephrased to read, “Like most current climate models, CESM does not simulate the observed Antarctic sea ice expansion. Our results show that this can be improved by manually correcting sea ice drift velocity biases.” (Lines 194-196)

[9] *Please check the References more carefully. Numerous ones have incomplete or erroneous bibliographic details. Some I notice were . . .*

Reply: We thank the reviewer for catching these typos. We have made the suggested corrections and carefully gone through the references in the revised manuscript.

References

- Bracegirdle, T. J., Hyder, P., and Holmes, C. R. (2018). CMIP5 diversity in southern westerly jet projections related to historical sea ice area: Strong link to strengthening and weak link to shift. *J. Clim.*, 31(1):195–211.
- Eisenman, I., Meier, W. N., and Norris, J. R. (2014). A spurious jump in the satellite record: has Antarctic sea ice expansion been overestimated? *Cryosphere*, 8(4):1289–1296.
- Ferreira, D., Marshall, J., Bitz, C. M., Solomon, S., and Plumb, A. (2015). Antarctic Ocean and sea ice response to ozone depletion: A two-time-scale problem. *J. Clim.*, 28(3):1206–1226.
- Fogt, R. L., Wovrosh, A. J., Langen, R. A., and Simmonds, I. (2012). The characteristic variability and connection to the underlying synoptic activity of the Amundsen-Bellinghousen Seas Low. *J. Geophys. Res.*, 117(D07111).
- Holland, M. M., Landrum, L., Kostov, Y., and Marshall, J. (2017). Sensitivity of Antarctic sea ice to the Southern Annular Mode in coupled climate models. *Clim. Dyn.*, 49:1813–1831.
- Irving, D. and Simmonds, I. (2015). A novel approach to diagnosing Southern Hemisphere planetary wave activity and its influence on regional climate variability. *J. Clim.*, 28(23):9041–9057.
- Irving, D. and Simmonds, I. (2016). A new method for identifying the Pacific–South American pattern and its influence on regional climate variability. *J. Clim.*, 29(17):6109–6125.
- Kostov, Y., Marshall, J., Hausmann, U., Armour, K. C., Ferreira, D., and Holland, M. M. (2017). Fast and slow responses of Southern Ocean sea surface temperature to SAM in coupled climate models. *Clim. Dyn.*, 48:1595–1609.
- Screen, J. A., Bracegirdle, T. J., and Simmonds, I. (2018). Polar climate change as manifest in atmospheric circulation. *Curr. Climate Change Rep.*, 4(4):383–395.
- Simmonds, I. (2015). Comparing and contrasting the behaviour of arctic and antarctic sea ice over the 35 year period 1979-2013. *Ann. Glaciol.*, 56(69):18–28.

REVIEWERS' COMMENTS

Reviewer #1 (Remarks to the Author):

Second Review: Sun and Eisenman

Thank you to the authors for addressing my questions. I am happy to recommend this paper for publication. I believe that it provides new insights into Antarctic sea ice trend biases in models and offers a new path forward to improve models. I only have two comments and a few minor edits.

The addition of figure S5 demonstrates that in 1992 the total SIA is smaller for the runs that include the OBSvi spin-up and those that do not. To what extent do you think that this overall decrease in climatological sea ice area over the 1960-1991 time period affects the model response to the OBSvi drift velocities from 1992 onwards? I am not suggesting the authors do any new experiments, but perhaps they could address this question.

Revisiting Figure 3, the clear signal of sea ice growth in the Weddell sea in the OBSvi experiments but not in the LENS seems like it might be due in part to the spatial resolution in CESM resulting in poor simulation of the orography of the peninsula region. To what extent do the authors think that model resolution might be affecting these sea ice drift biases?

Minor comments:

Line 50: why "Alternatively," here. It seems like you are still talking about changes in the surface winds and the related impacts in this paragraph.

Line 66: maybe you could say "mean wind-driven upwelling" to distinguish the Armour et al. argument from the Ferreira et al. argument.

Reviewer #2 (Remarks to the Author):

I thank the authors for their thoughtful and comprehensive response to my comments and those of my co-reviewer. The revised version works well, and has been expanded where appropriate and detail has been added where needed. Parts of the text have also been rephrased and the meanings are much clearer.

I am now pleased to be able to recommend acceptance of the paper.

Responses to Reviewer #1

We thank the reviewer for these helpful comments and suggestions. We have addressed the comments as follows in the revised manuscript (reviewer’s comments in italics):

[1] The addition of figure S5 demonstrates that in 1992 the total SIA is smaller for the runs that include the OBSvi spin-up and those that do not. To what extent do you think that this overall decrease in climatological sea ice area over the 1960-1991 time period affects the model response to the OBSvi drift velocities from 1992 onwards? I am not suggesting the authors do any new experiments, but perhaps they could address this question.

Reply: We appreciate the reviewer for raising this question. In response to this comment, we have added the following discussion to the revised manuscript (lines 246-251),

Due to the change in sea ice momentum forcing at 1960, the Antarctic sea ice area increases rapidly during the first few months such that the annual-mean ice area increases by around 1×10^6 km² in the first year (blue lines in Supplementary Figure 11a). The ice area then declines for a decade or so and then remains relatively constant during the following decade or so. After the 1960-1991 spin-up period, the ObsVi runs gain sea ice during 1992-2015.

We have also carried out additional simulations to further investigate the sensitivity to spin-up conditions, which are discussed in the revised supplemental material. The revised Figure S5 (now Supplementary Figure 11) and new Supplementary Table 1 summarize these simulations.

[2] Revisiting Figure 3, the clear signal of sea ice growth in the Weddell sea in the OBSvi experiments but not in the LENS seems like it might be due in part to the spatial resolution in CESM resulting in poor simulation of the orography of the peninsula region. To what extent do the authors think that model resolution might be affecting these sea ice drift biases?

Reply: We appreciate the reviewer raising this point. The coarse resolution is a possible reason for the sea ice drift biases in CESM. However, we cannot quantify how much this is contributing to the sea ice changes without carrying out a higher-resolution simulation. In response to this comment, we have acknowledged the potential impact of model resolution in the revised manuscript, which reads (lines 184-187), “The main candidates for explaining the remainder of the discrepancy between simulated and observed sea ice changes include model biases in the sea ice rheology, sea ice drag coefficients, and ocean surface currents, as well as ice velocity biases due to the coarse model resolution.”

[3] Line 50: why “Alternatively,” here. It seems like you are still talking about changes in the surface winds and the related impacts in this paragraph.

Reply: In response to this comment, we have re-structured the Introduction slightly to streamline this discussion in the revised manuscript.

[4] Line 66: maybe you could say “mean wind-driven upwelling” to distinguish the Armour et al. argument from the Ferreira et al. argument.

Reply: We appreciate the reviewer raising this suggestion. We have modified the text as suggested (line 67).